# Employees’ Entrepreneurial Dreams and Turnover Intention to Start-Up: The Moderating Role of Job Embeddedness

**DOI:** 10.3390/ijerph19159360

**Published:** 2022-07-30

**Authors:** Mingze Li, Jiaze Li, Xiaofang Chen

**Affiliations:** 1School of Management, Wuhan University of Technology, Wuhan 430070, China; mingze@whut.edu.cn; 2Research Institute of Digital Governance and Management Decision Innovation, Wuhan University of Technology, Wuhan 430070, China; 3School of Entrepreneurship, Wuhan University of Technology, Wuhan 430070, China; dumpling516@163.com

**Keywords:** entrepreneurial dreams, entrepreneurial self-efficacy, turnover intention to start-up, job embeddedness, identity theory

## Abstract

Many people have entrepreneurial dreams in mind, yet existing research has neglected to focus on this phenomenon. This paper introduces the concept of entrepreneurial dreams, constructs a model of the relationship between entrepreneurial dreams and turnover intention to start-up, based on identity theory and prospect theory, and empirically analyses the mechanism of the effect of entrepreneurial dreams on turnover intention to start-up. Through the analysis of data from two multi-provincial and multi-wave employee studies (Study 1 N = 198, Study 2 N = 227), the findings show that: (1) employees’ entrepreneurial dreams positively influence turnover intention to start-up; (2) employees’ entrepreneurial dreams can stimulate employees’ sense of entrepreneurial self-efficacy, thus positively influencing turnover intention to start-up; (3) job embeddedness plays a moderating role in the relationship between entrepreneurial self-efficacy and turnover intention to start-up, specifically, the higher the degree of job embeddedness, the weaker the effect of entrepreneurial self-efficacy on turnover intention to start-up; (4) job embeddedness moderates the indirect effect of entrepreneurial dreams on turnover intention to start-up through entrepreneurial self-efficacy, specifically, the higher the degree of job embeddedness, the weaker the indirect effect of entrepreneurial dreams on turnover intention to start-up through entrepreneurial self-efficacy. This study reveals the mediating role of employees’ entrepreneurial self-efficacy and the moderating role of job embeddedness in the influence of entrepreneurial dreams on employees’ turnover intention to start-up, which provides theoretical and practical references for relevant organizations.

## 1. Introduction

Many people aspire to be entrepreneurs, but the reality is that only a small percentage of those who pursue their dreams succeed. Most people must bury this dream in their hearts for the time being due to a lack of capital and contacts, a lack of management experience, the risk of starting a business with a low success rate, and a variety of practical difficulties. However, this unfulfilled dream does not vanish; it may remain in the mind and be integrated into the current job [1]. As a result, an interesting question to consider is whether this unfulfilled entrepreneurial dream has an impact on employees’ work.

The majority of the current entrepreneurship literature focuses on the various problems encountered by entrepreneurs or entrepreneurial teams who are in the process of starting their businesses or on their way to starting their businesses [2,3,4], while little attention has been paid to those who have dreams of starting their businesses but have not yet put them into practice, and the only relevant studies remain at the theoretical stage [5]. According to the existing literature, individuals’ understanding of the self is multidimensional and dynamic rather than simple and static, and in addition to the traditional focus on the strength of identity (e.g., identification), an increasing number of scholars are beginning to focus on the understanding of potential identities [6]. When relinquished professional identities are associated with unrealized values, individuals do not forget them, but attempt to retain them in their self-conception and find ways or opportunities to realize them [5]. Employees who consider and reflect on their unfinished path, their ‘other self’, can have an impact on their current work-life [7]; consequently, employees with entrepreneurial dreams in mind may seek ways to realize these unfinished dreams within the organization, and thus tend to leave to start their own businesses.

According to identity theory, the abandoned identity choice is not suppressed, but continues to influence people’s work lives [8], particularly when they are dissatisfied with their current work and life situation [9], and the ideal self repeatedly emerges and comes to the fore, prompting people to consider and imagine an alternative professional identity [8,10]. Comparing one’s current self to a better alternative self causes people to constantly substitute and internalize their imagined identity, and this imagined better alternative self leads people to believe that they could have been better in some way [8,11,12]. People’s dissatisfaction with their current career, job, and life grows stronger as they compare and imagine, as does their confidence in being competent in another professional identity, i.e., their self-efficacy. Entrepreneurial self-efficacy is an application of self-efficacy to the entrepreneurial domain and is a belief whereby individuals believe they are competent in different entrepreneurial roles and tasks [13]. Therefore, entrepreneurial self-efficacy is a key variable that influences the behavior of employees who have entrepreneurial dreams before they commit to action.

In addition to the confidence in being able to handle a new career, people usually consider the risky nature of entrepreneurship and ultimately make a choice based on a trade-off between risks and benefits. According to prospect theory, actors will use their perceived psychological utility to make judgments about risky decision-making behavior when making decisions. There are two stages in the human decision-making process: the early-editing stage, where relevant information is collected and collated, the initial analysis of prospects, and the subsequent evaluation decision stage [14]. Decision makers always tend to avoid harm and perceive the risk to a greater extent when making decisions, so in an uncertain environment, rational decision-makers will make decisions after constantly comparing gains and losses. Job embeddedness is defined as “the forces that prevent individuals from leaving their jobs” [15], where individuals and their families may be subject to various organizational or community constraints [16], which determine the cost of leaving. When employees are deeply ‘embedded’ in their work, leaving can be a significant sacrifice and loss, so job embeddedness is an important indicator for decision makers to assess the riskiness of entrepreneurial behavior and plays a moderating role in the relationship between entrepreneurial self-efficacy and turnover intention to start-up. Rational employees will only choose to leave their jobs if they expect the benefits to outweigh the risks.

Based on identity theory and prospect theory, this paper opens up the ‘black box’ of how employees with entrepreneurial dreams influence organizational behavior and it is the first empirical study to provide a deeper understanding of the psychological mechanisms of these employees. Although previous studies have focused on the various stages of entrepreneurial growth from the early to the later stages of entrepreneurship [17,18], less consideration has been given to those employees who are still in the stage of aspiring to start a business but have not yet started their entrepreneurial behavior. Therefore, this paper focuses on these employees and introduces the concept of entrepreneurial dreams as a way to explore the pathways that influence the behavior of this group of employees, contributing to the literature on theories and other aspects of entrepreneurship. Secondly, much of the research on employees within organizations has focused on the performance of employees [19,20,21] but has not considered the multiple identities that employees may have, the circumstances that may motivate them to work part-time or even leave their jobs, and the impact this may have on organizational behavior. This study enriches and extends the research findings on the effects of “abandoned professional identities” [5,10,21] by constructing a moderated mediating effect model and systematically analyzing the mechanisms and boundary conditions by which unrealized entrepreneurial identities affect employees’ actions, providing a useful addition to the related research. Our study also provides insights into practice, alerting organizations to pay more attention to the multiple identities of their employees and the dreams of their inner workforce. It is expected that the research herein will provide a general theoretical basis and broader ideas for future research.

## 2. Theory and Hypothesis

### 2.1. Employees’ Entrepreneurial Dreams and Turnover Intention to Start-Up

Entrepreneurship refers to the process of exploring and applying suitable opportunities and combining available resources to pursue a new business model [17]. Entrepreneurial dreams are created when individuals have a strong need to start a business [22]. A dream is defined as an emotion of need or an expectation for something that is missing, described as the need for something—a thing, a state, or a relationship [23,24]. As a combination of emotions, different kinds and levels of dreams that have an important impact on an individual’s psychological development, interpersonal relationships, and action tendencies [25,26,27]. Based on this, this paper introduces the concept of entrepreneurial dreams, which are defined as the emotions and desires of individuals regarding the need or expectation of entrepreneurial behavior. Unlike entrepreneurship, the entrepreneurial dream emphasizes the strong emotion of expecting to put into practice the entrepreneurial vision in one’s mind, a vision, and an expectation of entrepreneurial behavior, with aspiration at its core. Entrepreneurship, on the other hand, refers to the act of establishing a new business and becoming an entrepreneur [22] and is an important factor of production, with innovation at its core. In addition, the entrepreneurial dream is different from the entrepreneurial intention. Entrepreneurial intention is one of the indicators to predict entrepreneurial behavior. Entrepreneurial intention is defined as the belief that an individual plans to establish a new business and will consciously carry out these plans sometime in the future [28]. Entrepreneurial intention emphasizes the purposeful consciousness generated when people are guided by information related to entrepreneurship, which is the result of deep consideration. Only when people have confidence in entrepreneurial behavior will they have the will to engage in this behavior. The entrepreneurial dream emphasizes people’s initial vision for entrepreneurship. Regardless of whether they have various external and internal conditions conducive to the success of entrepreneurship, it will not affect people’s inner desires and expectations for entrepreneurship.

Due to the motivational properties of aspiration [24] and the tendency to approach action, the dream of entrepreneurship can thus inspire individuals to take action to approach their entrepreneurial goals, which may lead to the idea of leaving their current job [7]. Since there is a difference in purpose between employees leaving their jobs and choosing to start a business and leaving in general, a conceptual definition of this purpose-specific propensity to leave is needed. According to Mobley [29], the propensity to leave is defined as the employee’s idea or willingness to leave the organization. In this study, turnover intention to start-up is defined as the employee’s idea or willingness to leave the organization and start their business, a step that follows several other steps (the idea of leaving the organization, finding an entrepreneurial opportunity, evaluating entrepreneurial opportunities, etc.) and proceeds with the actual act of leaving to start a business. Based on identity theory, individuals will be potentially satisfied with an identity that is hidden within them but that is unrealized, which will motivate them to act towards achieving this identity [7]. Thus, employees with entrepreneurial dreams may further develop a turnover intention to start-up [7,8].

Firstly, employees with entrepreneurial dreams are more likely to experience dissatisfaction in their current job, which can lead to a tendency to leave [5,7,8]. Dreams are “possible fortunes and disasters” [30] and have an impact on individuals’ current behavioral decisions. Studies have shown that those who wish they had a different career are generally less satisfied and engaged with their current job and life [9]. Employees with entrepreneurial dreams have a strong imagination and desire for temporary unrealized entrepreneurial behavior, and this emotional component can lead to boredom, frustration, and even feelings of a lack of satisfaction with their current job status [5,7]. If an employee has tried to hold on to a renounced identity but has been frustrated, the desire for an entrepreneurial identity, rather than disappearing, becomes stronger, leading to feelings of regret and dissatisfaction [5,7,8]. Dissatisfaction with one’s current job is an important attitudinal variable influencing the propensity to leave [31,32,33]. Thus, employees with entrepreneurial dreams develop negative feelings of dissatisfaction with their current jobs and begin to look for ways to connect with their abandoned entrepreneurial identities as they contemplate and aspire to an unchosen entrepreneurial identity [5], resulting in a propensity to leave their jobs to start a business. Secondly, according to identity theory, people’s imagined selves tend to be better than their real selves and have higher expectations and aspirations for their alternative selves [7,8], which makes them more likely to develop the tendency to leave to start a business. Expectations of the alternative self are more likely to inspire disappointment in the real self [7,24], thus prompting employees to seize the opportunity to realize their dreams [8]. Therefore, having dreams of entrepreneurship tends to further motivate employees to give up their current identity and pursue their ideal self [5]. In addition, dreams are accompanied by imagination [29], as individuals will observe people who are currently successful in business and imagine that they would be successful if they started their own business. These imaginings can trigger happy emotions and encourage individuals to move in the direction of achieving the desired entrepreneurship [9,34]. Thus, employees with entrepreneurial dreams are more likely to leave their jobs with high expectations of their ideal entrepreneurial status. Finally, entrepreneurial dreams are unchosen career paths for individuals themselves, often linked to unrealized values, career goals, and meaning in life [8,9], and this intrinsic drive will prompt employees to seek opportunities to realize their dreams, which will trigger turnover intention to start-up. Based on the above analysis, this study proposes the following hypothesis.

**H1.** 
*Entrepreneurial dreams positively influence turnover intention to start-up.*


### 2.2. Mediating Mechanisms of Entrepreneurial Self-Efficacy

According to identity theory, the ‘alter ego’ is a component of the self-concept and influences people’s perceptions of themselves [7,8], and when individuals have the idea of starting their own businesses, their hypothetical entrepreneurial identity has an impact on their perceptions. Entrepreneurial self-efficacy refers to an individual’s belief that he or she is capable of performing different entrepreneurial roles and entrepreneurial tasks [17]. This study proposes that entrepreneurial dreams may have an impact on turnover intention to start-up through entrepreneurial self-efficacy.

Entrepreneurial dreams can influence feelings of entrepreneurial self-efficacy. On the one hand, individuals have experiences that can help them develop a sense of entrepreneurial self-efficacy [5,35]. According to identity theory, people with entrepreneurial dreams express their ideal entrepreneurial identity by engaging in entrepreneurial-related ideas and activities at work or in their free time to achieve an internal balance [7,8]. In this process, individuals acquire an increasing amount of experience related to their dream job [9], which facilitates an increase in entrepreneurial self-efficacy [36]; on the other hand, individuals with entrepreneurial dreams can accumulate their entrepreneurial self-efficacy through alternative experiences. Employees with entrepreneurial dreams actively explore and discover knowledge and information related to their dream industry, pay attention to and observe successful experiences within their dream industry [5,10], and begin to engage in a kind of conscious role-playing through imitation and imagination, where the achievements and behaviors of others convey information to the observer, increasing the belief that their own entrepreneurship can also be successful [7,37,38]. In addition, employees who dream of starting their own business may have goals and identity conflicts, and the emotion triggered by this identity conflict will stimulate the generation of entrepreneurial self-efficacy. Dreams are goal-oriented and can stimulate individuals to pursue them [7,9]. As employees have imposed rules and systems on their existing jobs that hinder them from pursuing entrepreneurship, employees with entrepreneurial dreams tend to conflate their real self with their ideal self, making the alter ego more prominent [5]. According to identity theory, the focus on unrealized identities tends to generate a strong desire for them [38,39], in which the self-concept is constantly reconstructed and new identities are internalized, contributing to the individual’s belief that he or she can be competent while acting as the alter ego [5,8], resulting in a sense of entrepreneurial self-efficacy, i.e., the individual believes that he or she is capable of gathering and implementing the necessary resources, skills, and abilities to be successful in future entrepreneurship [35].

Secondly, entrepreneurial self-efficacy is an important driver of turnover intention to start-up. Individuals tend to enjoy engaging in work with a high sense of self-efficacy and tend to avoid doing work with a low sense of self-efficacy [40]. Entrepreneurial self-efficacy reflects an individual’s positive evaluation of their entrepreneurial abilities and confidence in the success of their business, as well their certainty that they are competent as entrepreneurs. Studies have shown that people with high entrepreneurial self-efficacy will be more willing to make entrepreneurial decisions [40,41], and there is strong empirical support for the role of entrepreneurial self-efficacy in entrepreneurial intentions [42,43]. The potential to start a business depends heavily on an individual’s assessment of their own abilities; therefore, individuals with a high sense of entrepreneurial self-efficacy are more likely to be inclined to leave their current position to start a business.

Therefore, this paper argues that entrepreneurial dreams have a significantly positive effect on entrepreneurial self-efficacy and that the relationship between entrepreneurial dreams and turnover intention to start-up is mediated by entrepreneurial self-efficacy. Additionally, the following hypotheses are proposed in this study.

**H2.** 
*Entrepreneurial dreams are significantly and positively related to entrepreneurial self-efficacy.*


**H3.** 
*Entrepreneurial self-efficacy mediates the relationship between employees’ entrepreneurial dreams and their turnover intentions to start-up.*


### 2.3. Moderation Mechanisms for Job Embeddedness

Individuals are naturally associated with work-related contexts at their workplaces and are therefore embedded in the ‘web’ of the work environment [16], and leaving entrepreneurship requires individuals to leave their current position, so individuals need to consider the benefits and costs of leaving [16,44]. Job embeddedness is defined as “the combined cost of making an individual avoid leaving his or her job” [15] and is an important cost consideration for individuals leaving their jobs [16]. According to prospect theory, individuals make decisions based on the relative magnitude of gains and losses rather than the absolute magnitude. As a result, individuals with a high degree of job embeddedness have more tight social connections in the organization’s network of relationships and are embedded in the work–life network in a variety of combinations [16]. The more of these ties an individual has, the more likely he or she is to be ‘caught up’ in the existing job and the higher the cost of sacrifice in choosing to leave. Therefore, the degree of job embeddedness may moderate the relationship between employees’ entrepreneurial self-efficacy and their propensity to leave their jobs.

According to prospect theory, people usually make decisions not in terms of the absolute value of wealth but in terms of the relative magnitude of gains and losses, and the theory also suggests that the pain of loss is more sensitive than the joy of gain [14]. Although entrepreneurial self-efficacy drives entrepreneurial behavior, leaving the current work environment to choose a new career path is often accompanied by certain costs and sacrifices, and employees evaluate all the material and psychological losses caused by leaving [16]; therefore, turnover intention to start-up is the result of a combination of risk and benefit considerations. Employees with high job embeddedness, out of a closer relationship with their current organizational environment, have a weaker contribution of entrepreneurial self-efficacy to turnover intention to start-up: firstly, when job embeddedness is high, individuals are better matched to their current occupational environment [16] and their current occupation is better matched to their personal interests, values, and organization [16,28,45]. In this case, even if employees have a sense of entrepreneurial self-efficacy, the employees’ act of making the choice to start a business represents a renunciation of the above factors, and the uncertainty and complexity of entrepreneurship is a great challenge for employees, so they are less likely to choose to leave to start a business. Secondly, a high degree of job embeddedness means that individuals have stable interpersonal relationships that are difficult to disengage, such as mutually supportive colleagues [15,16]. In this case, even if employees are confident in starting their own business, they are aware of the huge costs of leaving to build a new network of people after starting their own business, thus reducing turnover intention to start-up. Prospect theory states that individuals’ perceptions of leaving are more focused on material and psychological losses rather than perceptions of benefits [16]. Therefore, the relationship between entrepreneurial self-efficacy and the propensity to leave is weaker for individuals with high job embeddedness. Conversely, for employees with low job embeddedness, on the one hand, they feel less matched in their current organization and are confident that they will be able to find a job that better matches their self-worth and expectations after leaving [46,47], and on the other hand, they believe that they will be able to handle various interpersonal relationships and build better relationship networks in their future organization [36,46]; therefore, their desire to leave is more motivated by their entrepreneurial self-efficacy. As a result, employees with low job embeddedness are more concerned with assessing their own capabilities than employees with high job embeddedness and are more likely to be encouraged by their positive evaluations and confidence levels, which in turn affects their behavioral motivation [36] and makes them more likely to leave the organization. Based on this, we suggest that there is a moderating effect of employee job embeddedness regarding the relationship between entrepreneurial self-efficacy and turnover intention to start-up. As a result, the following hypotheses are proposed.

**H4a.** 
*Job embeddedness moderates the relationship between entrepreneurial self-efficacy and turnover intention to start-up; specifically, the higher the degree of job embeddedness, the weaker the positive relationship between entrepreneurial self-efficacy and turnover intention to start-up, and the stronger the opposite.*


The relationship revealed by Hypothesis 3 and Hypothesis 4a further manifests itself as a mediated one; job embeddedness moderates the mediating effect of entrepreneurial self-efficacy on the relationship between employees’ entrepreneurial dreams and turnover intention to start-up. Specifically, when job embeddedness is higher, entrepreneurial self-efficacy has a weaker effect on turnover intention to start-up and the indirect effect of entrepreneurial dreams transmitted through entrepreneurial self-efficacy on turnover intention to start-up is weaker; conversely, the lower the job embeddedness, the stronger the relationship between entrepreneurial self-efficacy and turnover intention to start-up and the stronger the indirect effect of entrepreneurial dreams on turnover intention to start-up through entrepreneurial self-efficacy. Accordingly, we propose the that the mediating role hypothesis is moderated and show the conceptual model for this study in Figure 1.

**H4b.** 
*Job embeddedness moderates the indirect role of entrepreneurial self-efficacy between employees’ entrepreneurial dreams and turnover intention to start-up; the higher the degree of job embeddedness, the lower the indirect role of entrepreneurial self-efficacy between entrepreneurial dreams and turnover intention to start-up, and the stronger the inverse of this relationship.*


In summary, the conceptual model of this study is as follows:
Figure 1Conceptual Model.
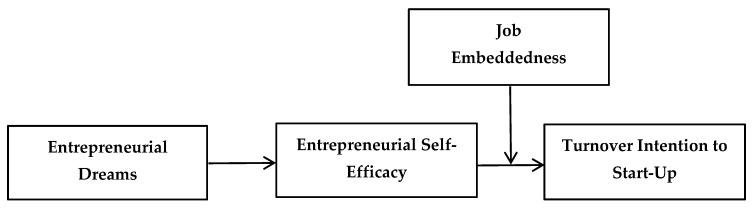



## 3. Research Methodology

### 3.1. Study 1 Method

#### 3.1.1. Sample and Procedure

This study used a questionnaire to collect data from enterprises and institutions in Beijing, Hubei, Hunan, Hebei, Jiangxi, Sichuan, Guangdong, Jiangsu, and Inner Mongolia Autonomous Regions in China. The online questionnaires were sent through the Questionnaire Star platform, a questionnaire research platform that has been proven to be effective in several studies [47,48]. To ensure the generalizability of the findings, this study tried to cover as many different industries and types of enterprises as possible in the research. The research sample was drawn from a variety of industries, including education, finance, real estate, restaurants, and telecommunications. An employee’s self-assessment was used as the key variable. The first survey covered employees’ entrepreneurial dreams and job embeddedness, while the second survey covered entrepreneurial self-efficacy and turnover intention to start-up. The second survey was only administered to employees who completed the first questionnaire with valid answers, and number matching was performed to ensure a valid match of data between the two time points. A total of 273 questionnaires were distributed in the first round and 263 questionnaires were returned, representing a return rate of 96.3%; 198 questionnaires were returned in the second round, representing a return rate of 72.5%. The final paired data of 198 employees was obtained for analysis. Among them, the age mainly concentrated in the 25 years old group and below, and secondly the 25–40 years old group, accounting for 40.9% and 35.9% respectively. The oldest participant was 59 years old. The proportion of males and females was 54% and 46%, showing a balanced gender ratio. The proportion of those with high school degrees or below was 14.1%, those with junior college degrees represented 22.7%, those with bachelor’s degrees represented 56.1%, and those with master’s degrees accounted for 7.1%. Years of working include new employees who had been on the job for a few months and old employees who had been working for 37 years. The total years of working with leaders was mainly reported to be 1–5 years, accounting for 80.3%. The shortest period of working with leaders was 3 months, and the longest was 17 years.

#### 3.1.2. Measures

The questionnaire includes variables such as entrepreneurial dreams, entrepreneurial self-efficacy, job embeddedness, and turnover intention to start-up. To ensure the validity of these scales in the Chinese context, the questionnaire was translated into Chinese using standard back translation procedures, and the meaning and presentation of the questionnaire items were discussed in a symposium format. All of the survey items were measured using a five-point Likert scale. The anchors for all items were “not at all” (1) to “fully conforming” (5). Appendix A presents the scale items for all our measures.

Entrepreneurial dreams. This study is based on the Abandonment of Identity Residency Scale developed by Burgess [21], which has been adapted to better suit the context of this paper. The scale consists of a total of 3 questions, including “I often think about my alter ego, such as becoming an entrepreneur”, “I often think about what my life would have been like if I had taken the ‘alternative path’ (such as becoming an entrepreneur)”, and “I often reflect on how things would have been different if I had become an entrepreneur”. The evaluation was carried out using a 5-point Likert scale, with different scores representing how often each description appears in the workplace (1 = hardly; 5 = always). The Cronbach’s alpha score scale was 0.881, indicating that the scale has good reliability.

Entrepreneurial Self-Efficacy. The entrepreneurial self-efficacy Scale developed by Wilson [49] was used in this study. The scale consists of 6 questions, including “I think I am capable of solving problems”, “I think I can make decisions”, “I think I have the ability to manage money”, “I think I am creative”, “I can get people to agree with me”, and “I can be a leader”. The Likert 5-point scale was used to evaluate the participants (1 = strongly disagree; 5 = strongly agree). The Cronbach’s alpha score was 0.881, indicating that the reliability of the scale was good.

Job embeddedness. The study used the Crossley [50] Job Embeddedness Scale, a seven-item scale, and selected items from this scale to measure the organizational dimensions of employees’ job embeddedness. These include “I feel a sense of belonging to this organization”, “It is difficult for me to leave this organization”, “I am so attracted to this organization that I cannot leave”, “I feel tied to this organization”, “I cannot leave the organization I work for at all”, “I am so connected to this organization”, and “It is easy for me to leave this organization”. The Likert 5-point rating scale was used for the evaluation (1 = strongly disagree; 5 = strongly agree). The Cronbach’s alpha score was 0.907, indicating the good reliability of the scale.

Turnover Intention to Start-Up. This study is based on the Turnover propensity Scale developed by Wayne [51], which was modified to better fit the scenario. The scale consists 5 questions, including “I will leave my company as soon as I can find a better entrepreneurial opportunity”, “I am actively looking for entrepreneurial opportunities”, “I am seriously considering quitting my job to start my own business”, “I often think about quitting my current job to start a business”, and “I think I will be working here in five years rather than starting a business”. The Likert 5-point rating scale was used for the evaluation (1 = strongly disagree; 5 = strongly agree). The Cronbach’s alpha score was 0.73, indicating that the scale has good reliability.

Control variables. Based on previous studies [33,52], the control variables were set as gender, age, education, years of working, and years of working with immediate supervisors.

#### 3.1.3. Results

To test the discriminant validity of the variables, a validation factor analysis was conducted on the model. As can be seen from Table 1, the fit of the four-factor model (χ^2^/df = 1.83, CFI = 0.941, TLI = 0.932, RMSEA = 0.065, SRMR = 0.065) was significantly better than the other alternative competing models, thereby indicating the good discriminant validity between the variables.

Table 2 specifies the means, standard deviations, and correlation coefficients for each of the main study variables. It is evident that employees’ entrepreneurial dreams are significantly and positively correlated with entrepreneurial self-efficacy, job embeddedness, and turnover intention to start-up (β = 0.267, *p* < 0.01; β = 0.26, *p* < 0.01; β = 0.195, *p* < 0.01). Entrepreneurial self-efficacy was significantly and positively correlated with job embeddedness (β = 0.165, *p* < 0.05), but the correlation with turnover intention to start-up was not significant. There was a negative correlation between job embeddedness and turnover intention to start-up, providing a preliminary basis for this paper to argue the research hypothesis.

Hypothesis 1 predicts that entrepreneurial dreams are positively related to the propensity to leave the workforce to start a business. This hypothesis was supported by a significant positive relationship between the two variables (Table 2: β = 0.195, *p* < 0.01) and was tested by a hierarchical regression after accounting for the control variables (see M4 of Table 3: β = 0.112, *p* < 0.05); thus, hypothesis 1 was tested. Hypothesis 2 suggests that entrepreneurial dreams are positively related to entrepreneurial self-efficacy, and we have confirmed a significant relationship between entrepreneurial dreams and entrepreneurial self-efficacy in M2 of Table 3 (β = 0.158, *p* < 0.001); thus, hypothesis 2 was tested. To test hypothesis 3, which stated that entrepreneurial self-efficacy would mediate the relationship between entrepreneurial dreams and turnover intention to start-up, we tested the regression model of entrepreneurial dreams and entrepreneurial self-efficacy as predictors of turnover intention to start-up in M5, but entrepreneurial self-efficacy was not a significant predictor of turnover intention to start-up (β = −0.022, *p* > 0.05). In addition, using the bootstrap method, the sample size was set to 5000, and the results showed an indirect effect of β = −0.003 with a confidence interval of [−0.035, 0.02] including 0. Therefore, the mediating effect was not valid and H3 was not verified.

Hypothesis 4a proposes that job embeddedness moderates the relationship between entrepreneurial self-efficacy and turnover intention to start-up. The interactive effect of job embeddedness is significant (see M7 of Table 3: β = −0.126, *p* < 0.05), and to more clearly show the moderating effect of job embeddedness, we plot the moderating effect, as shown by Figure 2. As shown in the figure, the effect of entrepreneurial self-efficacy on turnover intention to start-up is lower when job embeddedness is higher (β = −0.095, *p* > 0.05), while the effect of entrepreneurial self-efficacy on turnover intention to start-up is higher when job embeddedness is lower (β = 0.279, *p* < 0.05). Thus, hypothesis 4a was tested.

Finally, Hypothesis 4b predicts that job embeddedness moderates the mediating role of entrepreneurial self-efficacy between entrepreneurial dreams and turnover intention to start-up. The final model shown in Table 3 (M8) reveals the final step of a series of hierarchical regression models and the results of the combination of the mediating and moderating terms, confirming that the interaction holds when all the variables are taken into account. We used the bootstrapping method developed by Preacher et al. (2007) [53] to confirm the conditional indirect effects. The method estimates confidence intervals for effect sizes at one standard deviation above and below the mean of the moderating variable (see Table 4). As the 95% confidence interval was [−0.132, −0.006], excluding zero, the effect was considered significant and confirmed Hypothesis 4b.

#### 3.1.4. Discussion

In Study 1, we used a sample of 198 individuals to find that entrepreneurial dreams positively affect turnover intention to start-up (Hypothesis 1) and entrepreneurial self-efficacy (Hypothesis 2), that job embeddedness moderates the effect of entrepreneurial self-efficacy on turnover intention to start-up, and that different levels of job embeddedness influence the effect of entrepreneurial dreams on turnover intention to start-up through entrepreneurial self-efficacy (Hypothesis 4). Our theoretical model and hypotheses were partially supported.

### 3.2. Study 2 Method

#### 3.2.1. Sample and Procedure

The sample for Study 2 was drawn from multi-provincial enterprises and institutions, and three questionnaires were administered, each with an interval of one week and employee self-assessment as the key variable. The first survey included employees’ entrepreneurial dreams and job embeddedness, the second survey included entrepreneurial self-efficacy, and the third survey included propensity to leave and start a business. The questionnaires were measured using the same scales as in Study 1. Prior to the formal research, the research team first explained the purpose of the survey to the companies and stressed that the anonymity of the research and the confidentiality of the findings would be ensured. Once the formal questionnaire was compiled, the research team distributed and collected the questionnaires through the online questionnaire platform, Questionnaire Star. The participants were urged to complete the questionnaire with the support and assistance of the HR departments of the companies. The second survey was only distributed to those employees who completed the first questionnaire validly; similarly, the third survey was only distributed to those employees who completed the previous two questionnaires validly, and number matching was performed to ensure the effective matching of the data at the three time points.

A total of 278 questionnaires were distributed in the first round and 267 were returned, representing a 96% return rate; 256 questionnaires were returned in the second round, representing a 92.1% return rate; and 227 questionnaires were returned in the third round, representing an 81.7% return rate. The final paired data of the 227 employees was obtained for analysis. Among them, the age mainly concentrated in the 26–45 years old group, accounting for 56%. The proportion of males and females was 42.7% and 57.3%, showing a balanced gender ratio. The proportion of those with high school degrees or below was 15.4%, those with junior college degrees represented 26.4%, those with bachelor’s degrees represented 48%, and those with master’s degrees represented 10.1%. Years of working include new employees who had been employed at their current job for a few months and old employees who have been working for 35 years. The years of working with leaders were mainly concentrated in less than five years, accounting for 63.4%, and more than 30% had worked with their immediate leaders for more than five years.

#### 3.2.2. Results

In this study, a validated factor analysis was conducted on the four key variables “Entrepreneurial Dreams”, “entrepreneurial self-efficacy”, “turnover intention to start-up”, and “Job Embeddedness” using Mplus 7.4 to distinguish between the validity of the variables. The results of the four-factor model, three-factor model, two-factor model, and one-factor model are compared in Table 5. It is evident that the four-factor model fits better (χ^2^(183) = 402.6, RMSEA = 0.07, TLI = 0.94, CFI = 0.95) and significantly better than the other four models. This suggests that the four main variables in this study possess good discriminant validity among themselves.

Table 6 summarizes the results of the descriptive statistical analysis for each variable. As can be seen from Table 6, entrepreneurial dreams were significantly and positively correlated with both entrepreneurial self-efficacy and turnover intention to start-up (β = 0.394, *p* < 0.01; β = 0.33, *p* < 0.01), and there was a significant positive correlation between entrepreneurial self-efficacy and turnover intention to start-up (β= 0.303, *p* < 0.01). The above results were generally consistent with the study hypotheses and provided the initial support for the subsequent hypothesis testing.

Hypothesis 1 predicts that entrepreneurial dreams are positively associated with turnover intention to start-up. M4 in Table 7 shows the positive effect of entrepreneurial dreams on turnover intention to start-up (β = 0.222, *p* < 0.01); therefore, hypothesis 1 has been supported. Hypothesis 2 predicts that entrepreneurial dreams positively affect entrepreneurial self-efficacy, and the results of M2 in Table 7 show a positive relationship between entrepreneurial dreams and entrepreneurial self-efficacy (M2: β = 0.295, *p* < 0.01); therefore, hypothesis 2 has been supported.

Hypothesis 3 predicts that entrepreneurial self-efficacy mediates a positive indirect relationship between entrepreneurial dreams and turnover intention to start-up. The M5 results in Table 7 show that there is a significant positive relationship between entrepreneurial dreams and turnover intention to start-up (β= 0.156, *p* < 0.01) and that there is a significant positive relationship between entrepreneurial self-efficacy and turnover intention to start-up (β= 0.225, *p* < 0.01). The estimated mediating effect of entrepreneurial self-efficacy in the relationship between entrepreneurial dreams and turnover intention to start-up from a sample of 5000 bootstraps was 0.066, with *p* < 0.01, a 95% confidence interval of [0.022, 0.124], and did not include 0. Therefore, hypothesis 3 has been supported.

Hypothesis 4a proposes that job embeddedness moderates the relationship between entrepreneurial self-efficacy and turnover intention to start-up. The interactive effect of job embeddedness is significant (see M7 of Table 7: β = −0.134, *p* < 0.01). To show the moderating effect of job embeddedness more clearly, we plotted the moderating effect. As shown in Figure 3, when job embeddedness is higher, the effect of entrepreneurial self-efficacy on turnover intention to start-up is lower (β = 0.17, *p* > 0.05), whereas when job embeddedness is lower, the effect of entrepreneurial self-efficacy on turnover intention to start-up is higher (β = 0.515, *p* < 0.01). Therefore, hypothesis 4a was tested.

To test the mediated model with moderation, we then examined whether there were different indirect effects of entrepreneurial dreams on turnover intention to start-up at high and low levels of job embeddedness using a resample of 5000 bootstraps with entrepreneurial efficiency [53]. As shown in Table 8, the indirect effect was positive when the job embeddedness was low, with a significant indirect effect (0.125, 95% CI [0.074, 0.194]), with a confidence interval not containing 0. When the job embeddedness was high, the indirect effect was not significant (0.022, 95% CI [−0.046, 0.102]), with a confidence interval containing 0. Finally, the difference in the indirect effect at high and low job embedding levels was significant with a 95% confidence interval of [−0.196, −0.016]. In summary, hypothesis 4b was supported.

#### 3.2.3. Discussion

In Study 2, using a sample of 227, we again found that entrepreneurial dreams positively influenced turnover intention to start-up (Hypothesis 1) and entrepreneurial self-efficacy (Hypothesis 2), and that the relationship between entrepreneurial dreams and turnover intention to start-up was found to operate through entrepreneurial self-efficacy (Hypothesis 3), while job embeddedness moderated the mediating role of entrepreneurial self-efficacy between employees’ entrepreneurial dreams and turnover intention to start-up (Hypothesis 4). Therefore, our theoretical model and hypotheses are robustly supported.

## 4. Conclusions

### 4.1. Research Findings

Based on identity theory and prospect theory, this study constructed a theoretical model of employees with entrepreneurial dreams and the propensity to leave to start a business. Two survey studies were used to test the conceptual model and to explore in depth the process of the influence of entrepreneurial dreams on employees’ work. The main findings of Study 1 are as follows: firstly, employees’ entrepreneurial dreams can significantly and positively predict employees’ turnover intention to start-up, i.e., the stronger the entrepreneurial dreams, the more likely the employees are to choose to leave to start a business; secondly, entrepreneurial dreams have a significant positive impact on employees’ entrepreneurial self-efficacy, i.e., the stronger the entrepreneurial dreams, the more confident employees will be in their ability to achieve that dream and the higher their sense of entrepreneurial self-efficacy; thirdly, job embeddedness moderates the relationship between entrepreneurial self-efficacy and turnover intention to start-up, specifically, the higher the job embeddedness, the weaker the positive relationship between entrepreneurial self-efficacy and turnover intention to start-up. Job embeddedness also moderated the mediating role of entrepreneurial self-efficacy in the relationship between entrepreneurial dreams and turnover intention to start-up. Study 2, which replicated the results of Study 1, found that entrepreneurial self-efficacy mediated the positive relationship between entrepreneurial dreams and turnover intention to start-up.

### 4.2. Theoretical Contributions

The first attempt was to incorporate the entrepreneurial dream variable into an empirical study of organizational behavior. Most of the current literature on entrepreneurship focuses on entrepreneurs who are already on the entrepreneurial path and explores all the stages of growth from the start-up phase to the later stages of entrepreneurship [22,54]; however, no attention has been paid to the psychological and decision-making processes of those who aspire to be entrepreneurs before undertaking entrepreneurial activities. The study has not been able to explore the psychological and decision-making processes that precede entrepreneurial activity for those who aspire to it. In addition, although scholars such as Krueger [55] have researched entrepreneurial intentions more comprehensively, most of the research has been conducted from the perspective of objective resources [17,55] and has not focused on employees’ subjective desire to start a business. As an increasing number of people dream of starting their own business [56], it is important to explore how entrepreneurial dreams affect employees’ attitudes and behaviors. Based on identity theory, this study introduces the concept of “entrepreneurial dreams”, unveils the dark box of how employees’ entrepreneurial dreams affect their own organizational behavior, fills a gap in entrepreneurship research, provides empirical support for the application of identity theory to the field of entrepreneurship, and lays the foundation for future research on entrepreneurial dreams.

Secondly, this study enriches the relevant research on the factors influencing turnover intention to start-up. At present, there is a large amount of domestic and international literature on the factors influencing the propensity to leave [31,33], but the research on turnover intention to start-up is not yet complete. Firstly, there is no clear definition of turnover intention to start-up, and although research on the propensity to leave has made great progress [31,33,44], further exploration of the specific intentions of employees after incepting the idea of leaving is needed for a comprehensive understanding of the phenomenon of leaving; additionally, although a small number of scholars have focused on the entrepreneurial activities of employees leaving, the subjects of research are mostly technology-based employees [57,58], focusing on the background of these employees. The focus is on the background, work experience [59], and the identification of entrepreneurial opportunities [60] of these employees, but not on the psychological mechanisms and processes of employees with entrepreneurial dreams and the tendency to leave their jobs to start their own businesses. As the idea and phenomenon of quitting to start a business becomes more prevalent [56], there is an urgent need to enrich the research on exit-related entrepreneurial tendencies. This paper defines turnover intention to start-up, selects entrepreneurial self-efficacy as a mediating mechanism between entrepreneurial dreams and the propensity to leave to start a business, demonstrates the formation process of the propensity to leave to start a business, enriches the idea of studying the model of leaving entrepreneurship, gives further empirical support to the influencing factors of the propensity to leave to start a business, enriches the existing theory to a certain extent, and opens up new perspectives for future research.

Thirdly, based on prospect theory, this study proposes the moderating role of job embeddedness towards influencing turnover intention to start-up, and theoretically explains why the influence of entrepreneurial self-efficacy on turnover intention to start-up depends on the degree of employee job embeddedness. As a nascent theory in the study of employee turnover, job embeddedness has mostly been used as a mediating variable to examine the direct predictive effect on turnover [18,32,50]; however, turnover behavior is a decision made under multiple considerations [18,44]. It also reveals how the mediating effect of employees’ entrepreneurial self-efficacy on entrepreneurial dreams—turnover intention to start-up—decreases with an increasing job embeddedness. This finding not only adds to the research on job embeddedness as a variable, but also contributes to a deeper understanding of the relationship between entrepreneurial dreams, entrepreneurial self-efficacy, and turnover intention to start-up, with important implications for identity and prospect theory as well as organizational practice. In addition, the contextual conditions of this study help to provide a greater insight into the internal activities and considerations of employees when making the decision of whether to start a business or not.

### 4.3. Empirical Implications

Firstly, pay attention to the multiple identities that employees may have and the dreams in their hearts. In the past, the dreams of employees were often overlooked in management activities, and managers focused more on how to motivate employees to fulfill their current identities without paying attention to the other identities that may exist. As employees with entrepreneurial dreams tend to have a higher sense of entrepreneurial effectiveness and are more likely to leave their jobs to start their own businesses, as an organization manager, in the process of staffing and use, the manager should gain a deeper understanding of employees’ innermost thoughts, career plans, and personal goals, and pay full attention to their innermost dreams and provide them with opportunities to realize their dreams wherever possible through training, change management, encouraging internal entrepreneurship, etc. This will not only improve organizational performance, stimulate internal vitality, and promote continuous innovation, but also reduce brain fatigue to a certain extent.

Secondly, enhancing job embeddedness and strengthening the connection between employees and the organization is crucial. Job embeddedness is often used as a key predictor of turnover. The higher the job embeddedness, the stronger the connection between the employee and the organization and the greater the cost of breaking this connection, the greater the sacrifice and loss of the original benefits, and thus the less likely an employee is to leave. As a manager, if you want to reduce the turnover of talent within the organization, attract talent, and enhance the stickiness between talent and the organization, you need to strengthen the job embeddedness of employees and enhance the connection, trust, and dependency of employees towards the organization by strengthening the organizational embeddedness to tighten it and allow it to be more integrated. By providing a good working environment and welfare, interpersonal interaction, and resources inside and outside the organization, managers can also focus on material and emotional incentives; can make employees’ personal values, career growth goals, and development plans match the company’s mainstream culture, team style, and work requirements to the maximum extent; they can enhance work compatibility; and managers can make emotional investments in and care for employees’ lives, both on and off the job, so that the employees feel that they are integrated into the company. This will help employees to feel that they are integrated with the organization and the group they work for and that they feel safe, integrated, respected, and recognized in their current working environment, thereby increasing their job satisfaction and strengthening their attachment to the organization.

### 4.4. Limitations and Prospects

Although this study has enriched the mediating and moderating effects of the role of entrepreneurial dreams and turnover intention to start-up through an empirical research approach, there are some shortcomings in this study. Firstly, although we did two online studies (Study 1 N = 198 and Study 2 N = 227) to enhance the validity of the results, the outcome variable of this research was derived from self-assessed data on turnover intention to start-up, which may have some bias, and future studies could use objective data on exit-related entrepreneurial behavior from HR systems to further validate the reliability of this model. Secondly, although the test results show that the scale used in this study has good reliability and validity, the scales of entrepreneurial dreams and turnover intention to start-up in this study are based on the adaptation of other established scales, so future studies can test the reliability and validity of this scale with different sample sources. In addition, different positions of employees were not considered in the design of the questionnaire used in this study, which might have affected the results of the study; therefore, the effect of different positions can be further verified for improvement in future research. Finally, this study has focused only on the mediating role played by employees’ personal entrepreneurial self-efficacy from the perspective of identity theory; however, factors such as individual emotions and motivation may also play an important role in the relationship between entrepreneurial dreams and employees’ turnover intention to start-up, and future research could expand our study to explore other potential pathways of action. In addition, the background and policy conditions of establishing enterprises in different countries are different, which may also be factors affecting whether employees quit their jobs to start their own businesses. Interested scholars can consider the situation of different countries in future research.

## Figures and Tables

**Figure 2 ijerph-19-09360-f002:**
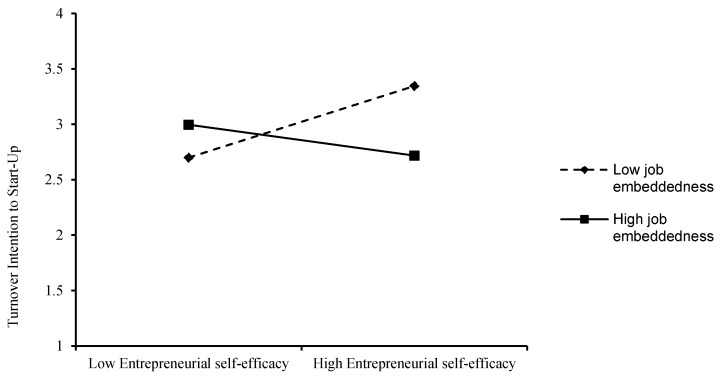
Interactive effects of Entrepreneurial Self-Efficacy and Job Embeddedness on Turnover Intention to Start-Up.

**Figure 3 ijerph-19-09360-f003:**
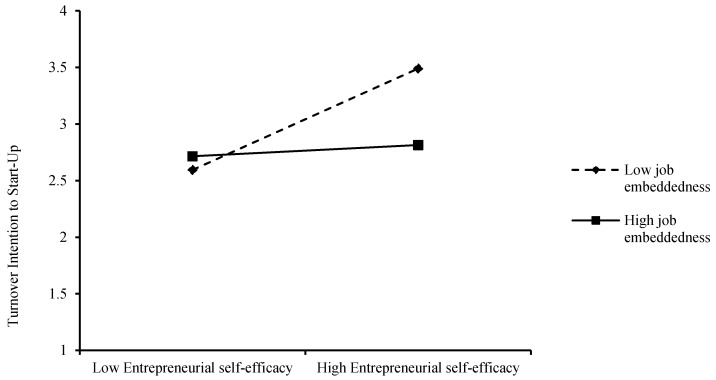
Diagram of the regulation effect.

**Table 1 ijerph-19-09360-t001:** Confirmatory factor analysis for discriminant validity.

Models	χ^2^/df	CFI	TLI	RMSEA	SRMR
Four-factor model	1.83	0.941	0.932	0.065	0.065
Three factors 1 (ED + ESE, JE, TISU)	2.63	0.759	0.730	0.123	0.130
Three factors 2 (ED, ESE + JE, TISU)	3.22	0.672	0.632	0.143	0.163
Three factors 3 (ED, ESE, JE + TISU)	2.98	0.708	0.672	0.135	0.128
Two-factor model	3.70	0.596	0.551	0.158	0.151
One-factor model	5.02	0.397	0.333	0.193	0.194

Note. ED = Entrepreneurial Dreams, ESE = Entrepreneurial Self-Efficacy, JE = Job Embeddedness, and TISU = Turnover Intention to Start-Up.

**Table 2 ijerph-19-09360-t002:** Descriptive Statistics: Means, Standard Deviations, and Correlations Among Variables.

Variable	*M*	*SD*	1	2	3	4
1. Entrepreneurial dreams	3.06	1.02				
2. Entrepreneurial Self-Efficacy	3.79	0.67	0.267 **			
3. Job Embeddedness	3.43	0.82	0.26 **	0.165 *		
4. Turnover Intention to Start-Up	2.59	0.72	0.195 **	0.054	−0.055	

Note: * *p* < 0.05; ** *p* < 0.01.

**Table 3 ijerph-19-09360-t003:** Hierarchical Regression.

	Entrepreneurial Self-Efficacy				Turnover Intention to Start-Up											
Predicator	M1		M2		M3		M4		M5		M6		M7		M8	
	β	SE	β	SE	β	SE	β	SE	β	SE	β	SE	β	SE	β	SE
gender	−0.162	0.094	−0.109	0.092	−0.251 *	0.103	−0.214 *	0.104	−0.216 *	0.104	−0.220 *	0.103	−0.229 *	0.103	−0.198	0.102
age	−0.103	0.081	−0.064	0.079	−0.11	0.089	−0.082	0.089	−0.084	0.089	−0.092	0.088	−0.129	0.088	−0.106	0.087
education	0.170 **	0.060	0.163 **	0.058	0.018	0.066	0.014	0.065	0.017	0.066	0.007	0.066	0.019	0.066	0.017	0.065
YW	0.179 *	0.073	0.171 *	0.071	0.068	0.081	0.063	0.080	0.067	0.081	0.087	0.081	0.075	0.081	0.086	0.080
YWL	−0.026	0.076	−0.036	0.074	−0.040	0.084	−0.046	0.083	−0.047	0.083	−0.032	0.083	−0.004	0.083	−0.009	0.082
ED			0.158 ***	0.045			0.112 *	0.050	0.116 *	0.052	0.141 *	0.053			0.133 *	0.052
ESE									−0.022	0.080	−0.014	0.079	0.092	0.081	0.045	0.0882
JE											−0.125	0.071	−0.083	0.068	−0.130	0.070
ESE × JE													−0.126 *	0.049	−0.119 *	0.048
R^2^	0.093		0.147		0.042		0.066		0.066		0.081		0.079		0.108	
ΔR^2^			0.054				0.024		0		0.015		−0.002		0.029	

Note: ED = Entrepreneurial Dreams; ESE = Entrepreneurial Self-Efficacy; JE = Job Embeddedness; YW = years of working; YWL= years of working with immediate supervisors. * *p* < 0. 05, ** *p* < 0. 01, *** *p* < 0.001.

**Table 4 ijerph-19-09360-t004:** Table of tests for mediating effects with moderation.

		Indirect Effect			
Moderator	Level	Effect Size	*SE*	BootLLCI	BootULCI
Job Embeddedness	Low (*M* − *SD*)	0.035	0.024	−0.001	0.095
	High (*M* + *SD*)	−0.021	0.019	−0.067	0.010
	Difference	−0.056	0.032	−0.132	−0.006

Note. high is +1 *SD* above the mean; low is −1 *SD* below the mean.

**Table 5 ijerph-19-09360-t005:** Results of confirmatory factor analysis.

Model	χ^2^	df	Δχ^2^	RMSEA	CFI	TLI	SRMR
four-factor model	402.609 **	183	—	0.073	0.948	0.940	0.064
three-factor model 1 (ESE + JE)	1671.908 **	186	1269.299	0.188	0.645	0.600	0.174
three-factor model 2 (ED + ESE)	921.987 **	186	519.378	0.132	0.824	0.802	0.106
two-factor model (ED + ESE, JE + TISU)	2301.372 **	188	1898.763	0.223	0.496	0.437	0.233
one-factor model	3001.923 **	189	2599.314	0.256	0.329	0.254	0.216

Note. ED = Entrepreneurial Dreams, ESE = Entrepreneurial Self-Efficacy, JE = Job Embeddedness, and TISU = Turnover Intention to Start-Up. ** *p* < 0. 01.

**Table 6 ijerph-19-09360-t006:** Descriptive statistical results and correlation coefficients of variables.

	*M*	*SD*	1	2	3	4	5	6	7	8	9
1. gender	1.570	0.496									
2. age	2.700	1.046	−0.040								
3. Education	3.480	0.979	−0.113	−0.28 **							
4. YW	3.360	1.500	−0.095	0.711 **	−0.170 *						
5. YWL	2.980	1.379	−0.076	0.653 **	−0.171 *	0.850 **					
6. ED	3.210	1.064	−0.048	−0.181 **	−0.020	−0.077	−0.050	(0.933)			
7. ESE	3.690	0.770	−0.113	0.079	−0.118	0.096	0.111	0.394 **	(0.931)		
8. TISU	2.830	0.770	−0.003	−0.154 *	−0.099	−0.125	−0.124	0.33 **	0.303 **	(0.805)	
9. JE	3.430	0.883	0.009	0.115	−0.150 *	0.118	0.138 *	0.22 **	0.244 **	−0.084	(0.905)

Note: YW = years of working; YWL= years of working with immediate supervisors; ED = Entrepreneurial Dreams; ESE = Entrepreneurial Self-Efficacy; JE = Job Embeddedness; TISU = Turnover Intention to Start-Up. * *p* < 0. 05, ** *p* < 0. 01.

**Table 7 ijerph-19-09360-t007:** Regression analysis table.

	Entrepreneurial Self-Efficacy				Turnover Intention to Start-Up											
Predicator	M1		M2		M3		M4		M5		M6		M7		M8	
	β	SE	β	SE	β	SE	β	SE	β	SE	β	SE	β	SE	β	SE
*Intercept*	4.204 **	0.330	2.952 **	0.356	3.748 **	0.329	2.803 **	0.368	2.139 **	0.410	2.533 **	0.419	2.73 **	0.200	2.462 **	0.413
gender	−0.188	0.103	−0.150	0.094	−0.047	0.102	−0.019	0.097	0.015	0.095	0.025	0.093	0.062	0.096	0.068	0.093
age	−0.016	0.071	0.071	0.066	−0.131	0.070	−0.065	0.068	−0.081	0.067	−0.072	0.065	−0.131 *	0.065	−0.078	0.064
education	−0.095	0.054	−0.0677	0.049	0.0125 *	0.054	−0.103 *	0.051	−0.088	0.050	−0.103 *	0.049	−0.114 *	0.050	−0.110 *	0.048
YW	−0.003	0.069	−0.008	0.063	0.080	0.069	0.004	0.066	0.066	0.064	0.005	0.063	0.008	0.063	0.004	0.061
YWL	0.057	0.070	0.034	0.064	−0.028	0.069	−0.046	0.066	−0.053	0.065	−0.044	0.063	−0.047	0.064	−0.040	0.062
ED			0.295 **	0.044			0.222 **	0.046	0.156 **	0.049	0.181 **	0.048			0.183 **	0.048
ESE									0.225 **	0.067	0.256 **	0.066	0.342 **	0.062	0.249 **	0.065
JE											−0.176 **	0.054	−0.106	0.056	−0.139 *	0.055
ESE × JE													−0.134 **	0.048	−0.136 **	0.047
R^2^	0.037		0.194 **		0.048		0.137 **		0.178 **		0.214 **		0.193 **		0.242 **	
ΔR^2^			0.157				0.089		0.041		0.036		−0.021		0.077	

Note: YW = years of working; YWL= years of working with immediate supervisors; ED = Entrepreneurial Dreams; ESE = Entrepreneurial Self-Efficacy; JE = Job Embeddedness; TISU = Turnover Intention to Start-Up. * *p* < 0. 05, ** *p* < 0. 01.

**Table 8 ijerph-19-09360-t008:** Conditional indirect effects of performing tension.

		Indirect Effect			
Moderator	Level	Effect Size	*SE*	BootLLCI	BootULCI
Job Embeddedness	Low (*M* − *SD*)	0.125 **	0.030	0.074	0.194
	High (*M* + *SD*)	0.022	0.037	−0.046	0.102
	Difference	−0.103 *	0.045	−0.196	−0.016

Note: * *p* < 0. 05, ** *p* < 0. 01.

## Data Availability

Not applicable.

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
