# Peer review of "Employees’ Entrepreneurial Dreams and Turnover Intention to Start-Up: The Moderating Role of Job Embeddedness"

_ijerph, 2022, doi:10.3390/ijerph19159360_

Round 1

Reviewer 1 Report

The paper provides good justifications for the study. Novelty was well established and the paper is methodologically sound.  Conclusions were based on the evidences and findings presented.  The authors need to include more latest literature published within the recent 5 years to ensure the currency  of the research gaps. 

Reviewer 2 Report

Comments to the Author

General suggestions or comments:

   my opinion, this study named " Employees' Entrepreneurial Dreams and Propensity to Leave to Start a Business: the Moderating Role of Job Embeddedness " now has a few new values. The author is required to make efforts to make some areas that could be strengthened.

Some specific points

1. Check the text and use of the phrase " entrepreneurial efficacy" Should it be " entrepreneurial self-efficacy "?

2. Moreover, the literature, although not very dated, should be improved by including more recent articles. 

3. Is the sample representative? 3.1.1 Sample and Procedure Please provide how the questionnaire was developed. Also, please consider providing a copy of the questionnaire as an Appendix or Supplementary material. In addition, it would be clearer if the authors provided a Table showing the demographic profiles of the respondents. 

4. The conclusions in the current article have few new values (most ones confirm what has already been said in literature.), and the suggestions are pretty obvious and based on limited evidence. For example, in the findings, is there also a difference in the sample's seniority in the company or its age in the willingness to leave to start a business?

5. There are many words in the text to be corrected, such as “ntre-preneurial” “posi-tively”, “em-beddedness” and so on. In addition, the reference format should also be adjusted to the form required by the journal.

Reviewer 3 Report

The article is interesting, but requires some major improvements.

First of all, I must write that I do not think that its issues fit well with scope of IJERPH Journal (focused on public health  and environmental policy, not entrepreneurship or psychology). The article is supposed to be a completely new approach to entrepreneurship research, it tries to present the role of dreams in entrepreneurial behaviors, therefore, in my opinion, it should be published in an important journal on this (E) subject. However, the editors of IJERPH should decide in this aspect.
I agree that the beginning of the article should interest the audience, but the first sentences sound too emotional and journalistic in style of tabloids not academic journals. There is a reference to the questionable quality of research on entrepreneurial dreams from one of the popular magazines (I do not know any serious scientific research that would confirm such high rates in market economies). So in my opinion this paragraph should be completly changed:

Entrepreneurship is the dream of many people. The well-known talent media "Ca-reer" magazine has launched a survey on the entrepreneurial as-pirations of the work-force. The survey results show that workplace people have a strong desire to start a busi-ness, 92.69% of the respondents said they have a dream or intention to start a business, but the fact is that the real dream will be put into practice is still a minority of people. The lack of cap-ital and contacts, the lack of management experience, the risk of starting a business with a low success rate ...... in a variety of practical difficulties, most people can only choose to temporarily buried in the heart of this dream, into the enterprise to do a "worker".
Moreover, this part of the article introduces differences between dream or intention without explaining these concepts (what is the difference between an enetrepreneurial dream and an entrepreneurial intention?).
This is important because later a sentence appears that can be considered completely untrue if we consider the "dream about entrepreneurship" to be the same or close to "entrepreneurial intentions". It is about the following sentence:
Although previous studies have focused on the various stages of en-trepreneurial growth from the early to the later stages of entrepreneurship [17,18], less consideration has been given to those employees who are still in the stage of aspiring to start a business but have not yet started their entrepreneurial behavior.

There are many publications about the entrepreneurial intentions of young people and students, also in emerging countries (see e.g. articles in EBER, https://eber.uek.krakow.pl/) for example in Indonesia, Tunisia, Oman, Poland (see e.g. https://eber.uek.krakow.pl/index.php/eber/article/view/142) in which students declare the intention to start their business, but they have not yet started their entrepreneurial behavior. So, this is rather dream... Therefore, I believe that the authors must clearly explain the differences between "dream" and "intention" in the context of entrepreneurship and their study. Just explaining what you mean "dreams" (on page 3) is not enough, especially since there is also the term "entrepreneurial efficacy" used in the study in very important place (H2).

The hypothesis should be better and more clearly formulated, eg. in H1 what is "the propensity to leave"? (to leave job/company? or to leave entrepeneurial intention? - we could find some statement on page 8, but it should be clear from the begining of the paper ). Also Fig. 1 should be more clear in this aspect. Moreover, Fig. 1 is not a theoretical model, but rather a conceptual one (it is not the same!).

The research sample was poorly described. Nothing is known about the respondents - what positions they occupy, what education they have (e.g. in business or management), which could have influenced their responses.

The context of running own business in China, which is not a capitalist country such as the countries of Western Europe or North America, is also unclear. Thus, I think that there are completely different legal and administrative conditions for starting a business in this countries vs. China. This can have a strong impact on anwers in the survey and the results of the study. Therefore, it should be described, because the readers from around the world may not even know if everyone in China can set up their own business, due to the large share of the authorities in the central control of the economic procesess. Therefore, it is necessary to briefly explain the domestic context of set up the business.

The discussion is quite weak, there are no references to the results of other studies. Even if the authors believe that their research is pioneering, they could refer to the results of other authors' research on entrepreneurial intentions.

Although the references list is very long, there are no references to other studies on the entrepreneurial behavior of people in China or other emerging (e.g. Asia) countries.

There are many not very understandable, too long sentences and minor language errors that need to be corrected, also in abstract (e.g. entrepre-neurial, en-trepreneurial or posi-tively with wrong hypens; the paper is full of hypens in wrong places in the middle of words; no proper commas in the end of page 2: ....other aspects of entrepreneurship Second, "First,.. Second...." in many places should be "Firstly,... Secondly,... etc. ).The article therefore requires a careful linguistic revision by the authors, it is advisable for a professional proofreader to do it also.

Reviewer 4 Report

the study was well-done and well-written

I noticed one minor writing concern for the journal staff to fix:

"Entrepreneurial dreams can influence feelings of entrepreneurial effi-cacy." -- the hyphen in efficacy should be removed. This issue occurs elsewhere in the article, thus a close proofreading is needed.

Since the R-square results of the hierarchical regress are low, but significant, it might help the readers if more detail is added to the discussion about what other possible variables might be included or switched out for future research

Round 2

Reviewer 2 Report

Good Job!